

# On the magnetic characteristics of magnetic holes in the solar wind between Mercury and Earth

Martin Volwerk[1], Charlotte Goetz[2], Ferdinand Plaschke[1], Tomas Karlsson[3], and Daniel Heyner[2]

[1]Space Research Institute, Austrian Academy of Sciences, Graz, Austria

[2]Institute for Geophysics and Extraterrestrial Physics, Technische Universität Braunschweig, Germany

[3]Department of Space and Plasma Physics, School of Electrical Engineering and Computer Science, Royal Institute of Technology, Stockholm, Sweden

*Correspondence to:* M. Volwerk, Space Research Institute, Austrian Academy of Sciences, Schmiedlstr. 6, 8042 Graz, Austria
(martin.volwerk@oeaw.ac.at)

**Abstract.** The occurrence rate of linear and pseudo magnetic holes has been determined during MESSENGER's cruise phase starting from Earth (2005) and arriving at Mercury (2011). It is shown that the occurrence rate of linear magnetic holes, defined as a maximum of $10°$ rotation of the magnetic field over the hole, slowly decreases from Mercury to Earth. The pseudo magnetic holes, defined as a rotation between $10°$ and $45°$ over the hole, have mostly a constant occurrence rate, with a slight increas in front of the Earth and a decrease around the Earth. The width and depth of these structures seem to strongly differ depending on whether they are inside or outside of Venus's orbit.

## 1 Introduction

The interplanetary magnetic field (IMF) shows various kinds of structures and discontinuities on different scales (see e.g., Burlaga, 1969; Burlaga et al., 1969) such as large scale section boundaries, coronal mass ejections (CMEs), corotational interaction regions (CIRs); middle scale reconnection exhausts; and small scale waves and turbulence. On the small scale side Turner et al. (1977) found that there were depressions in the IMF strength with $|B| < 1\gamma$ in an average field of $|B| \sim 6\gamma$, in the form of discrete "holes" ($1\gamma = 1$ nT). The rotation of the background magnetic field was very small from one side of the hole to the other. This set them apart from regular current sheets, and these so-called "linear holes" were assumed to be some diamagnetic structure created by inhomogeneities in the solar wind plasma, although the plasma data were of insufficient rate to determine that for sure. It is these linear holes, or magnetic holes (MHs) that are the topic of this paper.



The origin of these structures has been studied and e.g. Stevens and Kasper (2007) found that they occur mainly when the plasma-$\beta$ of the solar wind is high. This makes MHs related to structures that look similar, namely mirror mode (MM) waves, which are also characterized by magnetic depressions, usually in a "train" of structures, for high-$\beta$ plasmas with a temperature asymmetry $T_\perp > T_\parallel$ (Gary et al., 1993). Specifically, when $R_{\mathrm{SK}} > 1$ (Southwood and Kivelson, 1993) with

$$R_{\mathrm{SK}} = \frac{T_{p,\perp}/T_{p,\parallel}}{1 + 1/\beta_{p,\perp}}, \tag{1}$$

and

$$\beta_{p,\perp} = \frac{n_p k_B T_{p,\perp}}{B^2/2\mu_0}. \tag{2}$$

Interestingly, Stevens and Kasper (2007) found that the magnetic holes mainly occurred in MM stable environments. They argue that the non-linear development of MMs may result in MHs in
MM stable regions. Indeed, Hasegawa and Tsurutani (2011) proposed a turbulent diffusion model for the development of MMs (Bohm-like diffusion, Bohm et al., 1949), where the higher-frequencies of the structure diffuse out. Thereby smaller MMs will disappear, whilst larger MMs tend to grow in size as they are transported away by the plasma flow from their generation region. Using data from Venus Express at Venus and Giotto at comet 1P/Halley, Schmid et al. (2014) showed that indeed the
sizes of MMs increase when the spacecraft is further away from the assumed generation region.

      Buti et al. (2001) presented another generation mechanism based on presence of large-amplitude, right-handed Alfvén wave packages, observed in the solar wind to propagate at large angles with respect to the background magnetic field. Using hybrid simulations they show that these Alfvén wave packages develop into MHs in plasma regions where there is a high plasma-$\beta$, $T_e < T_i$ and
$T_{i,\perp} \neq T_{i,\parallel}$, through the creation of plasma inhomogeneities.

      In order to find out a possible origin region for and the development of MHs in the solar wind and their occurrence rate, the cruise phase of the MESSENGER (MErcury Surface, Space ENvironment, GEochemistry, and Ranging, Solomon et al., 2007) spacecraft on its way from the Earth to Mercury is studied. Several studies have discussed the occurrence rate of MHs (or MMs) in the solar wind.
Turner et al. (1977) found an occurrence rate of MHs of 1.5 per day near Earth using Explorer 43 (Imp I) data. Zhang et al. (2008) used Venus Express data near Venus to find an occurrence rate of MMs of 4.5 per day. Stevens and Kasper (2007) used Wind data to obtain a rate of 1 MH per 1.75 days. Briand et al. (2010) used both Cluster (between 2002 and 2005) and found 65 MH of which 45 were linear holes and also STEREO (between March 2007 and August 2009) and found there were
146 well defined structures of which 38% were linear MHs.

      Further out in the solar system MHs were found with the Ulysses mission, where Winterhalter et al. (2000) found at high solar latitude an occurrence rate of 5.2 per day, whereas Burlaga et al. (2007) used Voyager 1 data from 2006 when the spacecraft was located in the heliosheath and found an occurrence of 2 MH per day. Sperverslage et al. (2000) found various occurrence rates as they





studied the solar wind from 0.3 AU to 17 AU, with different space missions: From Helios 1 & 2
    1.7-2.2 per day, from Voyager 2 between 2 and 17 AU an average value of 0.2 per day, but with
    a decreasing trend from 0.2 per day between 2 and 4 AU to 0.1 per day beyond 11 AU. Naturally
    one needs to be careful comparing all these different occurrence rates as not all papers use the same
    criteria to determine the presence of MHs.

For MESSENGER's orbital phase Karlsson et al. (2016) studied isolated magnetic structures,
    which could be interpreted as magnetic holes. However, they found that there were both "neg-
    ative" and "positive" magnetic structures (i.e. decreases and increases in magnetic field strength
    respectively). Interestingly, the positive structures were only observed in the magnetosheath, and
    not present in the solar wind. It is not uncommon to find a combination of peaks and troughs in the
magnetosheath, where these structures develop from the mirror mode instability. Joy et al. (2006)
    showed how there was a development from troughs to peaks in the Jovian magnetosheath. How-
    ever, in the Hermean magnetosheath Karlsson et al. (2016) identify the positive structures as flux
    transfer events as many are associated with a bipolar field signature. The negative structures had
    $-0.5 \geq \Delta B/B \geq -1$ and a duration of $2 \text{ s} \leq \Delta t \leq 200 \text{ s}$, with no real difference between solar
wind and magnetosheath events.

## 2   The Data

This study is performed using the MESSENGER magnetometer data (Anderson et al., 2007) during
the cruise phase of the mission from Earth to Mercury (2005 - 2011). The data have a resolution of
1 s and are in heliocentric, cartesian J2000 coordinates.[1]

There is not continuous data for the cruise phase, as can be seen in Fig. 1 , where the different years
    of the cruise are plotted in different colours, and the circles show the location of where magnetic
    holes are observed, and Fig. 2, bottom panel, where the radial distance of MESSENGER from the
    Sun is plotted over time.

## 3   Magnetic Hole Finding Method

The magnetic field data are investigated for the presence of magnetic holes. In this paper the same
    method is used as in Plaschke et al. (2018). A short recap:

   – The background magnetic field strength is determined by a sliding window average over 300
     s: $B_{300}$;

   – The data are smoothed by a sliding window average over 2 s: $B_2$;

---

[1]Definition: The origin is at the center of the Sun with the fundamental plane in the plane of the Earth's equator. The
primary direction, the $x$-axis, points toward the vernal equinox of year 2000. With a right-handed convention specifying the
$y$-axis to point $90°$ to the east in the fundamental plane and the $z$-axis along the Earth's north polar axis.





– From the ratio time series $\Delta B/B_{300}$, lowest depressions are preselected that are separated by
        at least 300 s.

       – The total magnetic field strength $B_{300} > 2$ nT;

       – The depth of the structure $\Delta B/B_{300} = (B_{300} - B_2)/B_{300} > 0.5$;

       Out of the seven years of data this results into an identification of 8124 structures, of which the
full width at half maximum (FWHM, in seconds) is determined and the depth in $\Delta B/B_{300}$. Using
       the location of MESSENGER in interplanetary space at a resolution of one hour, we can determine
       an estimate of the dwelling time of the spacecraft at a certain radial distance from the Sun.

       One more restriction needs to be put onto the MH events: the rotation of the magnetic field should
       be small over the structure. In order to check this, the average magnetic field is determined by
the time interval before the structure, $\mathbf{B}_{bef}$ during $[t_0 - 2\delta t \ \ t_0 - \delta t]$, and after the structure, $\mathbf{B}_{aft}$
       during $[t_0 + \delta t \ \ t_0 + 2\delta t]$, where $t_0$ is the location of the structure and $\delta t$ is the FWHM of the
       structure. The rotation angle is then determined by $\Theta = \angle(\mathbf{B}_{bef}, \mathbf{B}_{aft})$. In Fig. 3 an example of
       a MH structure with $\Theta < 10°$ is shown and in Fig. 4 an example of a structure with $\Theta > 170°$ is
       shown which indicates a current sheet (CS) instead of an MH, which is also clear from the magnetic
field components shown.

## 4   Results

       The solar wind magnetic field varies with distance from the Sun, decreasing in strength the further
       from the Sun, in an approximately $R^{-1}$ dependence. In Fig. 5 a 2D histogram of the mean magnetic
       field strength is shown as a function of $R$, for the whole cruise mission.
In this study, unlike studies during the orbital phase of MESSENGER (e.g. Karlsson et al., 2016),
       there is no need to discuss the influence of solar activity, as can be seen in Fig. 2, top panel. The blue
       line shows the monthy averaged sunspot number and the red line shows the cruise phase, which is
       all the way in very low solar activity. Although, during the orbital phase of the mission, there is little
       dependence between the number of observed magnetic holes and the sunspot number (*Karlsson et*
*al., 2019, paper in progress*).

       The occurrence rate of the MHs as a function of radial distance from the Sun is studied first.
       Therefore, the region $0.3 \leq R \leq 1.1$ AU is binned into bins of 0.05 AU. For each bin the number
       of magnetic holes and the dwelling time was determined, after which the ratio of the two gives the
       occurrence rate per hour. The histogram is given in Fig. 6, where the data are also split up into
rotational bins: $\Theta \leq 10°, 10° < \Theta \leq 45°, 45° < \Theta \leq 90°, 90° < \Theta \leq 135°, 135° < \Theta \leq 180°$.

       On average there is a $21.9 \pm 5.5$ % chance to observe a structure, in one hour, which relates
       to $\sim 5.6$ per day, although there are variations in the bins. In Fig. 6 the colour coding shows the



various rotation angle ranges defined above and also the results from other studies have been added translated into a rate in %/hr.

### 4.1 linear MHs

For the linear MHs (LMHs), i.e. $\Theta \leq 10°$ the average occurrence rate is 9.0 %/hr with a standard deviation of 3.5 %/hr, which means $\sim 2.2$ MHs per day. The standard deviation also means that the decrease of LMH events (3.3 and 4.1 %/hr) near Earth is significant at near $2\sigma$ deviation. Nevertheless, this is not the only effect, there is a clear slow decrease in the occurrence rate from Mercury to Earth and then an increase again, and thus the mean may not be meaningful.

For the LMHs a 2D histogram of the occurrence rate of the width (FWHM) and the depth of the LMHs is shown in Fig. 7. The width of the LMHs inside the orbit of Venus, at $\sim 0.72$ AU shows that they have mainly a width between 15 and 30 seconds, whereas outside the width is spread between 15 and 100 seconds.

The physical size of the LMHs inside Venus's orbit, assuming a solar wind $v_{\mathrm{sw}} = 350$ km/s, would then be $\sim 5000 \leq \mathcal{W} \leq 10000$ km. With an assumed magnetic field strength near Mercury of $\sim 10$ nT and $v_{\perp} = v_{\mathrm{sw}}$, this would correspond to $\sim 13 - 28$ proton gyro radii. Outside Venus's orbit the width can be larger, up to $\mathcal{W} \sim 35000$ km, and with a magnetic field strength near Earth of $\sim 5$ nT this would correspond to $\sim 7 - 48$ proton gyro radii.

The depths of the LMHs as shown in Fig. 7 seem to have a different distribution too inside and outside Venus's orbit. Inside the depth is spread up to 0.85 with the highest counts in the lower depth bins. The distribution outside seems to be more erratic.

### 4.2 "Pseudo" MHs

"Pseudo' MHs (PMHs) in this paper are defined as the structures with a slightly larger rotation of the magnetic field, i.e. $10° < \Theta \leq 45°$, the orange part in Fig. 6. The average occurrence rate of these structures is 7.9 %/hr with a standard deviation of 2.6%/hr, which means $\sim 1.9$ PMHs per day. As in the LMHs the occurrence rate near Earth 4.7 and 4.1 %/hr the deviation from the average is almost $1.5\sigma$. In this case, the occurrence rate is relatively constant with an increase between 0.8 and 0.9 AU and a decrease between 0.95 and 1.05 AU.

In Fig. 8 the same 2D histograms for the width and the depth of the PMHs are shown as for the LMHs. It is clear that the spread in the width is larger for these structures, but again, comparing between inside and outside Venus's orbit there is a difference, with inside the highest occurrence rates are between 15 and 60 seconds, whereas outside there is a more constant occurrence rate between 15 and 150 seconds, slowly decreasing to between 15 and 100 seconds.

For the depth the highest occurrence rates inside are found below 0.7. Outside of Venus's orbit the highest occurrence rates are concentrated below 0.85, but the width seems to drop off between Venus and Earth





### 5 Magnetic Field Strength

The mirror instability, Eq. (1), is dependent on $\beta_{p,\perp}$ and thus on the background magnetic field
strength and the plasma parameters. In the case of MESSENGER there are no directly available
plasma data for the cruise phase and therefore, the events are only studied as a function of the
background magnetic field $B_{300}$ for both the width and the depth of the structures.

The LMHs, Fig. 9, show that the width is mainly concentrated below 40 s for all field strengths.
However a trend can be observed that the wider LMHs happen for stronger background fields, see
e.g. the points between 120 and 160 s around $B \sim 20$ nT. Similarly, there seems to be a broadening
of the distribution of the depth of the linear MHs as a function of $B$. In the 2D histogram an
exponetial fit was made to the approximate boundary of the high-occurrence rate, where for $B = 1$
nT a depth of 0.5 was assumed. This resulted in the green line with $D(B) = a \exp\{-bB\} + c$ with
$a = -0.53 \pm 0.07$, $b = 0.17 \pm 0.06$ nT$^{-1}$ and $c = 0.93 \pm 0.06$.

Qu et al. (2007) discussed the gyro-kinetic model of the MM instability and find growth rates on
the order of $\gamma_{\mathrm{MM}} \propto 10^{-2}\Omega_{\mathrm{i}}$, where $\Omega_{\mathrm{i}}$ is the ion cyclotron frequency. Based on only linear growth
of the structures this would indicate that for stronger $B$ stronger MMs can be expected, however it
should be expected that non-linear behaviour sets in at some point.

Similarly for the PMHs, Fig. 10 shows that the spread in width is broader than for the LMHs,
but again with a more larger width for stronger $B$, although there is also a broader distribution for
lower magnetic field strengths up to 5 nT. The distribution of the depth is also broader for the lower
magnetic field strengths, however, the green exponential curve seems to fit the strongest occurrence
rates also rather well.

### 6 Discussion

There are few papers that discuss the development of the MHs as they are transported by the solar
wind in interplanetary space. Sperverslage et al. (2000) used the Helios 1 and 2 data to search for
MHs in the region between Mercury and Earth, similar to what was done in this paper with the
MESSENGER data. Estimated from their figure 7 it can be found that the averaged over 0.2 AU
width of the MHs slightly increases when moving away from the Sun, from $\sim 13$ s at $R \sim 0.3$ AU
to $\sim 18$ s at $R \sim 0.8$ AU. This trend cannot be confirmed from the results in this paper, shown
in Fig. 7 left panel. However, a trend to longer structures does exist in the PMHs shown in Fig. 8
through a slight broadening of the counts between 0.3 and 0.7 AU.

The occurrence rate near the Earth (bins 0.95 - 1.00 and 1.00 - 1.05 AU) are significantly lower
with 3.3 and 4.1 %/hr, respectively, than the average occurrence rate between Mercury and the Earth,
which was found to be $\sim 9.0$ %/hr with a standard deviation of $\sigma \approx 3.5$ %/hr. Thereby a difference
of $\sim 1.5\sigma$ for these two bins. Xiao et al. (2014) used Cluster data to find LMH (trains) in the solar
wind during 2001 - 2009, and found over these years an occurrence rate of $1.8 \pm 0.8$ per day, or





7.5 ± 3.3 %/hr (see Xiao et al., 2014, Table 1). For the years 2005 and 2006, when MESSENGER was still relatively near Earth, the occurrence rates for Cluster were 1.6 and 1.2 per day (6.6 and 5.0

190   %/hr), respectively. However the definition of an MH is slightly different with $B_{\mathrm{min}}/B \leq 0.75$ and a maximal rotation of $\Theta \leq 15°$.

However, there is also the gradual decrease of the occurrence rate from Mercury to Earth. This can have two origins. Firstly, there can be a constant number of LMHs, but as the solar wind transports them outward they get "diluted" by $R^{-1}$. Fitting the occurrence rate with an power function $aR^b$

gives $a = 5.9 \pm 1.1$ and $b = 0.82 \pm 0.18$ with $\mathcal{R}^2 = 0.81$. Secondly, there could be a decay of the LMHs with an exponential drop $a \exp\{bR\}$ for which the fit returns $a = 24 \pm 5$ and $b = -1.5 \pm 0.4$ with $\mathcal{R}^2 = 0.88$. This behaviour does not hold for the PMHs.

The data in this paper show that the LMHs do not change much in size as they travel from Mercury to Venus, but the distribution of their depths seem to slightly narrow towards smaller values. This

means that there is statistically no development of these structures. Their characteristics change, however, outside of Venus's orbit, where the width of the LMHs have a broader range, whereas the depth does not seem to be much different.

For the structures with a larger rotation, the PMHs, the minimum width increases as they move away from the Sun. Just outside of Venus's orbit the distribution of the widths is broader, and then

narrows towards smaller sizes. The depth also seems to change just outside of Venus's orbit to reduce similarly as the width.

In general, there is a slight increase in MH width for both types between Mercury and Venus. This would be expected from the Bohm-type diffusion (Hasegawa and Tsurutani, 2011), where the size of MM structures is described by:

$$\lambda(L) = \rho_{\mathrm{p},0} \left( 1 + \frac{\omega_{\mathrm{c,p}} L}{32 u} \right)^{1/2}, \tag{3}$$

with $\lambda$ the scale size, $L$ the traveled distance of the structure with convection velocity $u$ and $\rho_{\mathrm{p}}$ and $\omega_{\mathrm{p,c}}$ the proton gyro radius and frequency respectively. The term $\rho_{\mathrm{p},0} = 9\rho_{\mathrm{p}}$ comes from the maximum growth rate for MM waves. Taking average values for the solar wind ($u \approx 400$ km/s, $B \approx 5$ nT) results in $\lambda(L) \approx 47\rho_{\mathrm{p},0}$ for a distance $L = 0.4$ AU between Mercury and Venus. This

kind of growth is not observed in Figs. 7 and 8, where there is at most a factor 2-3 in growth. This means that, when MHs behave similarly as MMs the structures have to be created at all locations between Mercury and Venus, and grow over a distance of $80 \times 10^3 - 210 \times 10^3$ km and decay again. Joy et al. (2006) state that the decay (or collapse) of these structures is stochastic, that there are different decay times for different structures. Dedicated numerical simulations should cast light

on this issue.

The distribution of the width beyond Venus are quite different, with much wider structures and lesser depth. A broad range of widths up to 180 s occurs at 0.85 AU and then the range decreases. Similarly, the depth of the structures is broader at 0.85 AU and narrows further out to lesser values.





Based only on the distributions of the width and depth, this would indicate that these are different
structures in the solar wind as compared to the region inside Venus's orbit. The lower occurrence
rate of MHs near Earth is currently being investigated using the Magnetospheric MultiScale (MMS
Burch et al., 2016) mission. Also the cruise phase of the BepiColombo mission (Anselmi and Scoon,
2001) should be used to investigate these structures.

It should be noted that the occurrence rate, due to the 300 s distance selection criterion, is a lower
limit. In the case of a wave train only the deepest hole is selected, and the train is seen as a single
event. However Winterhalter et al. (1995) found that wave trains mainly appear in mirror-mode
unstable regions, whereas in the stable regions solitary magnetic holes are found. They offer the
explanation that when MMs are created and move into a MM stable region the weakest MMs diffuse
and only the strongest develops into an MH. Therefore, counting a wave train as one event seems
to be defensible. For events separated like shown in Fig. 3 there is a slight miscount. Checking for
smaller distances between events shows an error of approximately 10% in counts.

## 7 Conclusions

Magnetic holes are ubiquitous in the solar wind. Using the MESSENGER magnetometer data during
the cruise phase between Earth and Mercury, the occurrence rate, width and depth of linear and
pseudo magnetic holes (LMH and PMH) was determined during solar minimum conditions. A
similar study should be done during the cruise phase of the BepiColombo mission. (Anselmi and
Scoon, 2001), The main results are:

– There is a slow decrease in the LMH occurrence rate from Mercury to Earth from $\sim 14\%/hr$
to $\sim 4\%/hr$, whereas for the PMHs it is rather constant with a increase between 0.8 and 0.9
AU, followed by a decrease between 0.95 and 1.1 AU.

– The difference between the LMHs and PMHs occurrence rate over the observation interval
basically rules out the "dilution" or "parametric decay" of the structures. Also, the significant
difference between the widths of the MHs inside and outside of Venus's orbit rules this out.

– Inside Venus's orbit there is a narrow range of widths between $\sim 4$ and $\sim 30$ s, whereas
outside the range is between $\sim 4$ and $\sim 180$ s. Assuming that the MHs also show Bohm-like
diffusion argues for a constant creation and diffusion and (stochastic) decay of these structures
between Mercury and Venus.

– Outside of Venus's orbit the MHs are wider and their depth seems to decrease with increasing
$R$. This would indicate that the MHs outside of Venus's orbit are different structures from
those inside.

*Acknowledgements.* The MESSENGER data were obtained from NASA's PDS (https://pds.nasa.gov/). The
sunspot numbers were obtained from SILSO (http://www.sidc.be/silso/home).





The work by CG was financially supported by the German Ministerium für Wirtschaft und Energie and the Deutsches Zentrum für Luft- und Raumfahrt under contract 50 QP 1401.

D. Heyner was supported by the German Ministerium für Wirtschaft und Energie and the German Zentrum für Luft- und Raumfahrt under contract 50 QW 1501





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





**Figure Captions**

**Fig. 1.** The cruise phase of MESSENGER projected onto the $XY_{J2000}$-plane with different colours for different years and the location of the second Venus flyby (V2) and the three Mercury flybys (M1/2/3). The circles denote the location at which magnetic holes are observed.

**Fig. 2.** Top: Monthly average sunspot number in blue and the MESSENGER cruise phase in red. Bottom: the radial distance of MESSENGER from the Sun during the cruise phase. The periods where no data are available are clearly visible.

**Fig. 3.** Two magnetic holes on 4 February 2008 between 1125 and 1130 UT with rotation over the structure $\Theta < 10°$ and depth $\Delta B/B_{300} > 0.8$. Top panel shows the magnetic field magnitude full resolution (blue), $B_{300}$ (red) and $B_{10}$ (green). Note that this time interval would count as one event even though 2 holes can be seen, which have a separation of $\sim 2$ min.

**Fig. 4.** A current sheet event on 23 August 2008 at 1607 UT with rotation over the structure $\Theta > 170°$ and depth $\Delta B/B_{300} > 0.8$. The magenta circle in the top panel was identified as a MH candidate by the search program.

**Fig. 5.** 2D histogram of the mean magnetic field as a function of $R$ for the whole cruise mission. The bins are 0.05 AU in $R$ and 0.5 nT for $B$.

**Fig. 6.** Histogram of the occurrence rate of MHs as a function of radial distance and colour coded after the rotational bins as given in the legend. For each category the single count statistical error is determined and plotted as an errorbar. The magenta asterisks in the bottom panel show the occurrence rate near Venus (Zhang et al., 2008, Z) and Earth (Turner et al., 1977, T), (Winterhalter et al., 2000, W), (Stevens and Kasper, 2007, SK) and (Xiao et al., 2014, X). The two dashed magenta lines are the average Helios occurrence rate (Sperverslage et al., 2000, SP)

**Fig. 7.** 2D histogram of occurrence rate of: Left - the width (in seconds) and Right - the depth of the LMHs as a function of radial distance from the Sun.

**Fig. 8.** 2D histogram of occurrence rate of: Left - the width (in seconds) and Right - the depth of the PMHs as a function of radial distance from the Sun.





**Fig. 9.** 2D histogram of occurrence rate of: Left - the width (in seconds) and Right - the depth of the linear MHs as a function of the background magnetic field strength. The dashed green line is an exponential fit to the approximate upper boundary of the depth of the structures.

**Fig. 10.** 2D histogram of occurrence rate of: Left - the width (in seconds) and Right - the depth of the pseudo MHs as a function of the background magnetic field strength.



**Figures**

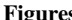

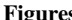

**Fig. 1.** The cruise phase of MESSENGER projected onto the $XY_{J2000}$-plane with different colours for different years and the location of the second Venus flyby (V2) and the three Mercury flybys (M1/2/3). The circles denote the location at which magnetic holes are observed.



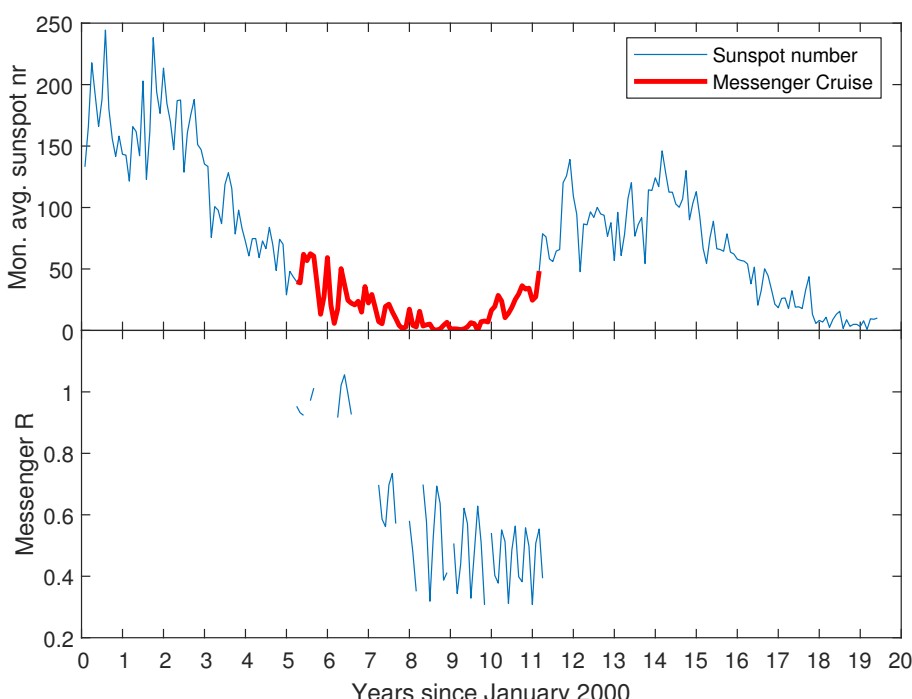

**Fig. 2.** Top: Monthly average sunspot number in blue and the MESSENGER cruise phase in red. Bottom: the radial distance of MESSENGER from the Sun during the cruise phase. The periods where no data are available are clearly visible.





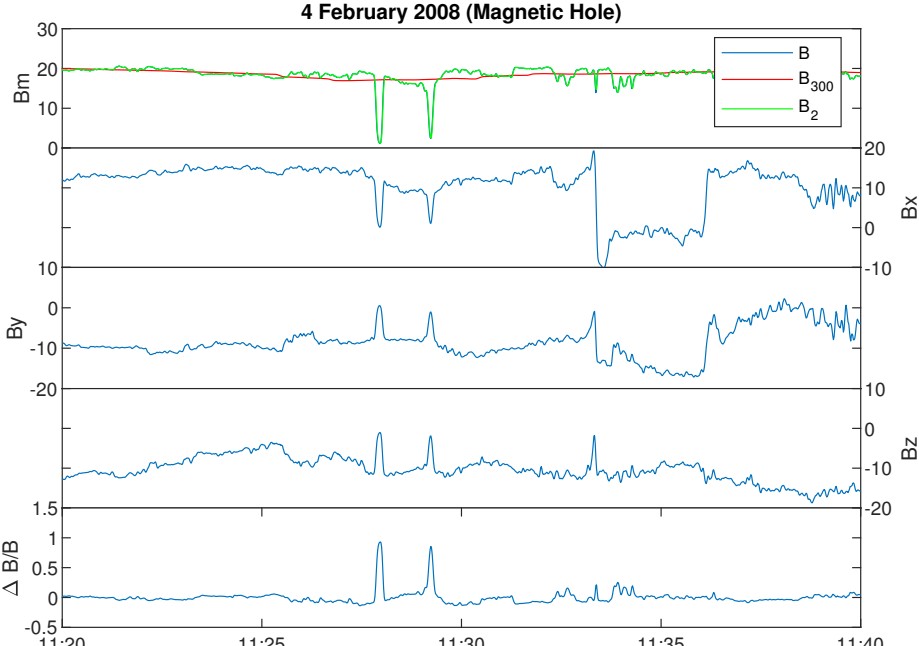

**Fig. 3.** Two magnetic holes on 4 February 2008 between 1125 and 1130 UT with rotation over the structure $\Theta < 10°$ and depth $\Delta B / B_{300} > 0.8$. Top panel shows the magnetic field magnitude full resolution (blue), $B_{300}$ (red) and $B_{10}$ (green). Note that this time interval would count as one event even though 2 holes can be seen, which have a separation of $\sim 2$ min.





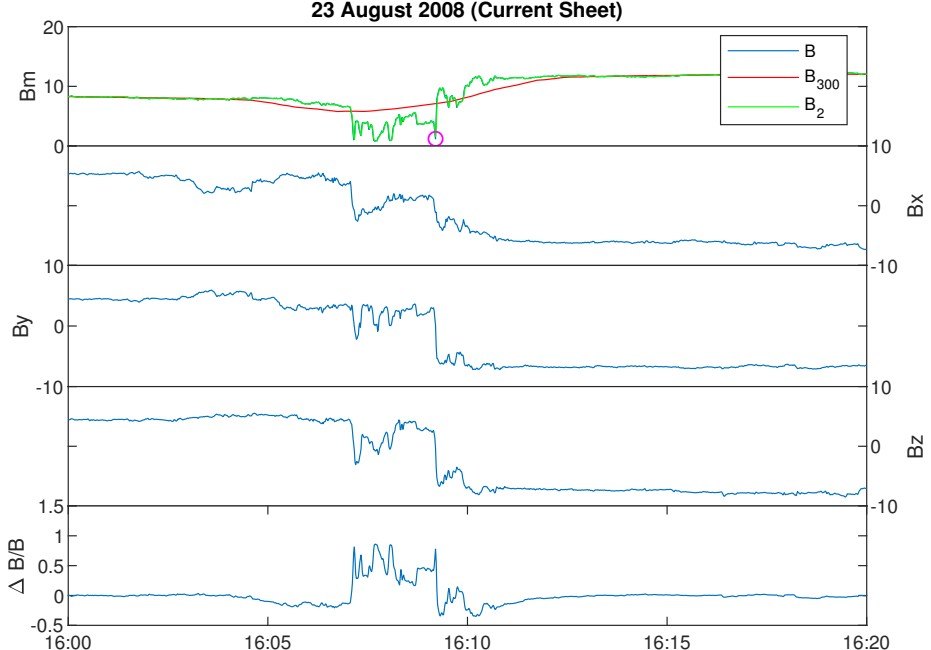

**Fig. 4.** A current sheet event on 23 August 2008 at 1607 UT with rotation over the structure $\Theta > 170°$ and depth $\Delta B/B_{300} > 0.8$. The magenta circle in the top panel was identified as a MH candidate by the search program.



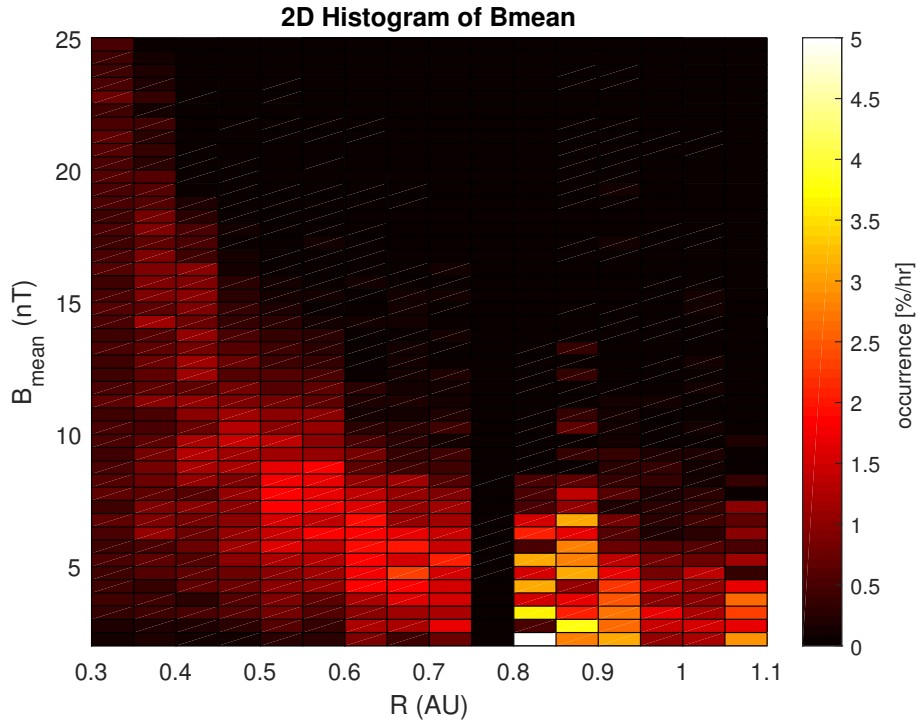

**Fig. 5.** 2D histogram of the mean magnetic field as a function of $R$ for the whole cruise mission. The bins are 0.05 AU in $R$ and 0.5 nT for $B$.

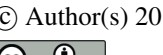



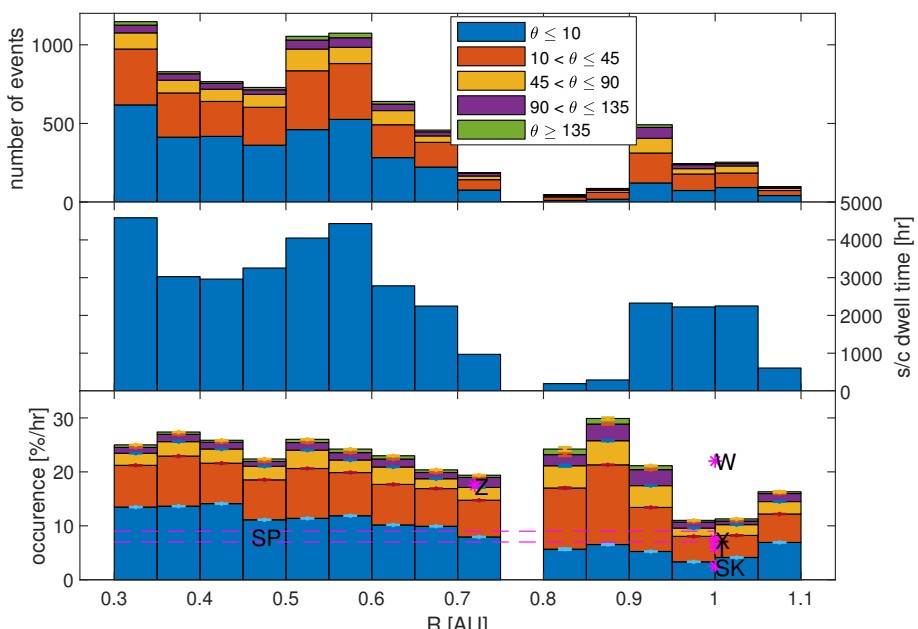

**Fig. 6.** Histogram of the occurrence rate of MHs as a function of radial distance and colour coded after the rotational bins as given in the legend. For each category the single count statistical error is determined and plotted as an errorbar. The magenta asterisks in the bottom panel show the occurrence rate near Venus (Zhang et al., 2008, Z) and Earth (Turner et al., 1977, T), (Winterhalter et al., 2000, W), (Stevens and Kasper, 2007, SK) and (Xiao et al., 2014, X). The two dashed magenta lines are the average Helios occurrence rate (Sperverslage et al., 2000, SP)

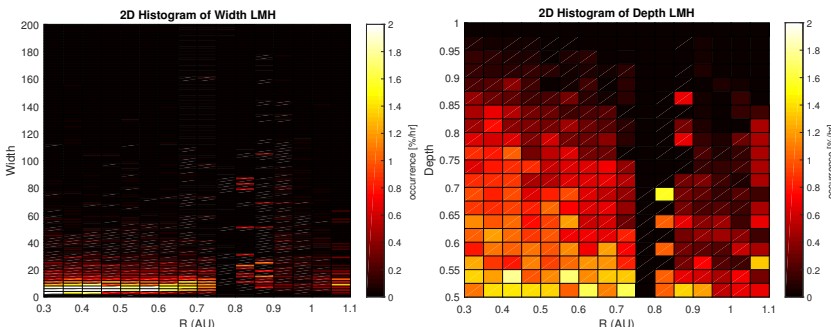

**Fig. 7.** 2D histogram of occurrence rate of: Left - the width (in seconds) and Right - the depth of the LMHs as a function of radial distance from the Sun.





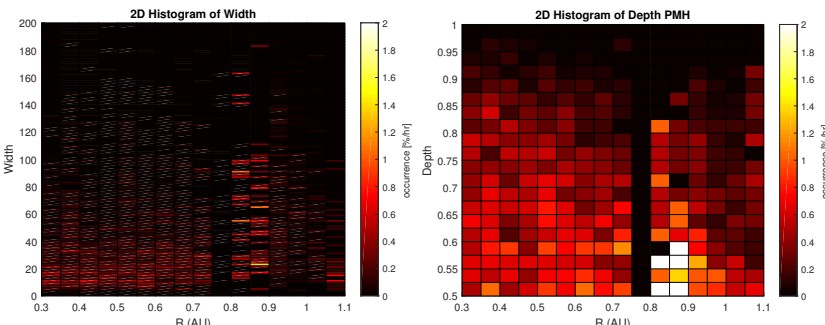

**Fig. 8.** 2D histogram of occurrence rate of: Left - the width (in seconds) and Right - the depth of the PMHs as a function of radial distance from the Sun.

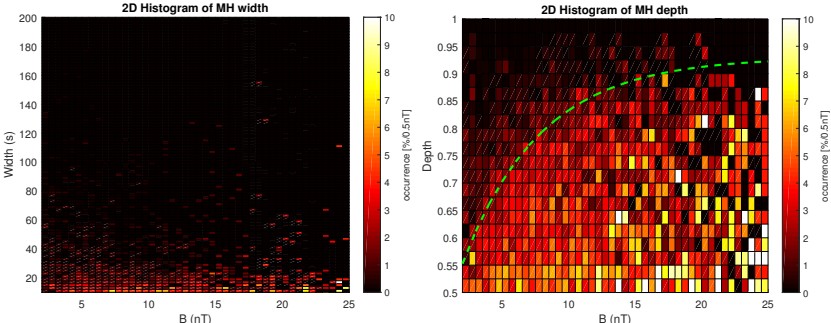

**Fig. 9.** 2D histogram of occurrence rate of: Left - the width (in seconds) and Right - the depth of the linear MHs as a function of the background magnetic field strength. The dashed green line is an exponential fit to the approximate upper boundary of the depth of the structures.

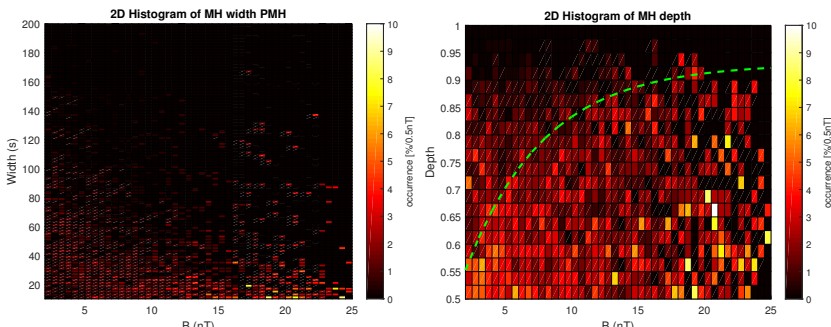

**Fig. 10.** 2D histogram of occurrence rate of: Left - the width (in seconds) and Right - the depth of the pseudo MHs as a function of the background magnetic field strength.