# Peer review of "On the magnetic characteristics of magnetic holes in the solar wind between Mercury and Venus"

_Annales Geophysicae, 2019_

## Referee Comment (RC1) · Anonymous Referee #1 · 13 Oct 2019

Magnetic Holes in the solar wind are a narrow, but unsolved classical problem in the micro-structure of interplanetary space plasmas. The MESSENGER magnetic field data set between 1 and 0.3 AU is unique and highly appropriate data set for the study of magnetic holes. The authors do an excellent job of summarizing what is know about magnetic holes and how they appear to evolve with distance from the Sun. The new MESSENGER results presented here strongly support the conclusion that the orbit of Venus, i.e. 0.7 AU, is an important break-oint with a marked change in how the holes evolve with distance inside and outside of this point. The paper is very well written, the analyses are sound, the figures are all necessary and well done, and the conclusions are well supported and significant. I recommend that the current version

of the manuscript be accepted for publication.

---

## Referee Comment (RC2) · Anonymous Referee #2 · 24 Oct 2019

General Comments

================

The article by Volwerk et al. discusses the observations of magnetic holes in the interplanetary magnetic field observed using MESSENGER cruise phase data. The authors present a detailed examination of the magnetic holes observed within the inner solar system, making important conclusions about the evolution and decay of such events with radial distance from the Sun. Their work is presented clearly and is discussed fluently; the figures and sections are organised in a logical way which suits the explanation in the paper. Other work done in this field is well cited, and their own

contribution is clearly stated. This is a valuable contribution, and as stated in their conclusion, could eventually be extended using data from the BepiColombo mission during its cruise phase.

Specific Comments

==================

line 80: What is the advantage of using this particular method over the others which may have been used in past studies of these events?

line 85: Why do they have to be separated by 300 s? Figure 3 shows two magnetic holes, but it only counts as a single one due to this criterion - are many events missed out because of this?

Figures 3 and 4: A little more description of these figures (i.e. what parameters are plotted in each panel) should be included both in the caption and where they are first mentioned in the text (lines ∼95-100).

line 125: Is the increase in occurrence past Earth meaningful? The gradient of the decrease in occurrence moving away from Mercury's orbit could possibly be quantified.

line 129: There are very few events in the bins past the orbit of Venus - this should be commented upon here, I think you have a fairly comparable number of events and width bins in the left panel Figure 7 which makes the statistics harder to trust than those from between Mercury and Venus.

lines 135-135: Again, could the erratic distribution outside of the orbit of Venus be attributed to a small number of events?

Section 4.2: It's a bit difficult to visualise any trends in the PMHs using the stacked histogram in Figure 6 - maybe this either warrants its own figure, or at least a separate panel on Figure 6

Figure 8: While I agree that the events are much more spread out in terms of their width

outside the orbit of Venus, the two bins within 0.8 < R < 0.9 contain relatively few events observed during relatively little dwell time - one width bin in particular (at around 25 s) appears to dictate the range of your colour bar, would it be worth adjusting the colour bar range to make the other columns easier to see? It would improve the visibility of the more statistically significant columns at the expense of the lesser ones.

line 160: How many events fall into the significantly higher width bins mentioned here? It could be argued that the spread of the widths at higher (>15 nT) field strengths are more difficult to gauge because there appears to be relatively fewer events spread across many width bins, whereas those with B < 15 nT seem to exhibit a fairly contin-uous distribution. The hypothesis that there is a trend in the widths of the LMHs could be made more obvious in some way using the B < 15 nT section of the left panel of Figure 9, for example by overlaying some lines indicating percentiles, or the standard deviation.

line 178-180: Are these numbers from the top panel of figure 7 of Sperverslage et al., 2000? If so, it looks to me like there is actually an increase in width from ∼6 or 7 s at 0.3 AU, to about 9 or 10 s at ∼1 AU (granted - the figure is difficult to read with much precision). To me, it looks like this trend is visible in in your figure 7 - maybe using the mean or mode of your radial bins, you might be able to confirm this by plotting them over the current figure 7, alongside the Sperverslage et al. numbers.

Technical Corrections

=====================

line 3-5: There are some weird issues with the spacing between characters on these lines in the PDF (I'm not sure if these are actually typos - they are probably just prob-lems with the PDF itself) e.g. "a s a m aximum o f 1 0 âŮę r otation o f"

line 6: "increas" -> "increase"

Figure 2: Please mention the units in the bottom panel on the axis label and/or the

caption.

Figure 5: It would be nice if the $R^{-1}$ dependence was shown over the top of the histogram.

Figure 6: Please make the dashed magenta lines a little more obvious (maybe another colour, or thicker lines). The magenta asterisks could also be made a little more obvious by changing the colour.

line 142: This sentence doesn't make seem to make much sense - in particular the bit "As in the LMHs the occurrence rate near Earth 4.7 and 4.1 %/hr", could you clarify the meaning of this please?

line 162: "exponetial" -> "exponential"

---

## Author Comment (AC1) · 18 Nov 2019

We would like to thank the referee for their kind words and their recommendation. Based on information obtained about the quality of the data between Venus and Earth, the analysis of the data has now been limited to the region between Mercury and Venus.

---

## Author Comment (AC2) · 18 Nov 2019

We would like to thank the referee for their careful reading of our paper. Also their kind comments on our work are appreciated. Below we will answer the specific comments of the referee.

First we need to comment on a change in the data that has been analysed. The MESSENGER data before 7 March 2007 cannot be used for data analysis, due to unrecoverable problems in calibration. Therefore the paper is not limiting its analysis to the region between Mercury and Venus.

line 80: What is the advantage of using this particular method over the others which may have been used in past studies of these events?

MV -> There is no specific advantage to the method we used, it is just that the analysis program was already written for a previous paper (Plaschke et al., 2018) and it is a very fast way of determining the holes. In principle this is the same method that e.g. Zhang et al. (2008) used.

line 85: Why do they have to be separated by 300 s? Figure 3 shows two magnetic holes, but it only counts as a single one due to this criterion - are many events missed out because of this?

MV -> The 300 s were chosen as to not identify wave trains of holes (or rather mirror mode waves) as all separate events. It is expected that the MM evolves into MH in MM stable regions in the solar wind. Indeed, Figure 3 shows two holes separated by less than 300 s and thus we are missing here one count. This is commented on at lines 98 – 103. However, at the end of the discussion, lines 262 – 269, specifically deals with this, where we state that by checking the data for smaller distances we can derive an error of ∼10% on the determination of the number of MHs.

Figures 3 and 4: A little more description of these figures (i.e. what parameters are plotted in each panel) should be included both in the caption and where they are first mentioned in the text (lines ∼95-100).

MV -> We have added a better description of the figures in the text, lines 98 – 106, and in the captions of figures 3 and 4. The changed parts are in red in the new version of the paper, in both the text and in the captions.

line 125: Is the increase in occurrence past Earth meaningful? The gradient of the decrease in occurrence moving away from Mercury's orbit could possibly be quantified.

MV -> The data beyond the orbit of Venus have now been taken out, therefore, the increase beyond the Earth is no longer part of the paper. A linear gradient can also

be made although we see no process that would have a linear decay of the MHs in the solar wind. For both the region between Mercury and Venus (Fmv) and for the complete data set (Fall) a linear fit was made to the occurrence rate as a function of R, which results in the following: Fmv = 18. − 12 R with R = [0.325 : 0.05 : 0.725]. We have extended the discussion about this in the text. A new figure (11) has been added showing the change in occurrence rate for LMHs and PMHs with different kind of fits, the power law and exponential decay and a linear fit.

line 129: There are very few events in the bins past the orbit of Venus - this should be commented upon here, I think you have a fairly comparable number of events and width bins in the left panel Figure 7 which makes the statistics harder to trust than those from between Mercury and Venus. lines 135-135: Again, could the erratic distribution outside of the orbit of Venus be attributed to a small number of events?

MV -> This is no longer current due to the deletion of the data beyond the orbit of Venus.

Section 4.2: It's a bit difficult to visualise any trends in the PMHs using the stacked histogram in Figure 6 - maybe this either warrants its own figure, or at least a separate panel on Figure 6

MV -> The referee is correct, this is difficult to see the way it was presented. We have decided not to stack the histograms anymore, but use side-by-side, which makes the determination of the PMHs trend easier.

Figure 8: While I agree that the events are much more spread out in terms of their width outside the orbit of Venus, the two bins within 0.8 < R < 0.9 contain relatively few events observed during relatively little dwell time - one width bin in particular (at around 25 s) appears to dictate the range of your colour bar, would it be worth adjusting the colour bar range to make the other columns easier to see? It would improve the visibility of the more statistically significant columns at the expense of the lesser ones.

MV -> The colour bars and the figures have been changed, as the data are only studied between Mercury and Venus.

line 160: How many events fall into the significantly higher width bins mentioned here? It could be argued that the spread of the widths at higher (>15 nT) field strengths are more difficult to gauge because there appears to be relatively fewer events spread across many width bins, whereas those with B < 15 nT seem to exhibit a fairly continuous distribution. The hypothesis that there is a trend in the widths of the LMHs could be made more obvious in some way using the B < 15 nT section of the left panel of Figure 9, for example by overlaying some lines indicating percentiles, or the standard deviation.

MV -> The referee is right, this part is unclear and we have re-analysed the data, where different procentiles of the width of the MHs are overlayed on the 2D histogram. Shown are now the 25, 50 75 and 97.5 procentiles. This section of the paper, lines 170 - 177, has been rewritten in view of the new versions of figures 9 and 10, left panels.

line 178-180: Are these numbers from the top panel of figure 7 of Sperverslage et al., 2000? If so, it looks to me like there is actually an increase in width from 6 or 7 s at 0.3 AU, to about 9 or 10 s at 1 AU (granted - the figure is difficult to read with much precision). To me, it looks like this trend is visible in in your figure 7 - maybe using the mean or mode of your radial bins, you might be able to confirm this by plotting them over the current figure 7, alongside the Sperverslage et al. numbers.

MV -> Indeed, this can be further detailed in the paper. In Figs. 7 and 8 we have added the location of the maximum growth rate in each radial bin by a green line and in Fig. 7 we have added the slope from Sperverslage et al. (2000) in a cyan line. There is a reasonable match between the two lines in Fig. 7. The text at lines 197 - 201 has been updated, as well as the captions of Figs. 7 and 8.

Technical Corrections ===================== line 3-5: There are some weird issues with the spacing between characters on these lines in the PDF (I'm not sure if

these are actually typos - they are probably just problems with the PDF itself) e.g. "a s a m aximum o f 1 0 â°U ËŻe r otation o f"

MV -> This must be a problem of the pfd as generated by AG, as we cannot reproduce this on our local pc. In the downloaded pdf from AG we see the same strange spacing.

line 6: "increas" -> "increase"

MV -> corrected

Figure 2: Please mention the units in the bottom panel on the axis label and/or the caption.

MV -> added

Figure 5: It would be nice if the $RËE{-1}$ dependence was shown over the top of the histogram.

MV -> A 1/R curve has been added to the figure.

Figure 6: Please make the dashed magenta lines a little more obvious (maybe another colour, or thicker lines). The magenta asterisks could also be made a little more obvious by changing the colour.

MV -> with the change of the bottom panel this does not seem a problem anymore

line 142: This sentence doesn't make seem to make much sense - in particular the bit "As in the LMHs the occurrence rate near Earth 4.7 and 4.1 %/hr", could you clarify the meaning of this please?

MV -> This sentence has been changed for clarify its meaning.

line 162: "exponetial" -> "exponential"

MV -> corrected
* * *
[Figure]

2019.